# UNCERTAINTY-AWARE DEEP VIDEO COMPRESSION WITH ENSEMBLES

## ABSTRACT

Deep learning-based video compression is a challenging task and many previous state-of-the-art learning-based video codecs use optical flows to exploit the temporal correlation between successive frames and then compress the residual error. Although these two-stage models are end-to-end optimized, errors in the intermediate errors are propagated to later stages and would harm the overall performance. In this work, we investigate the inherent uncertainty in these intermediate predictions and present an ensemble-based video compression model to capture the predictive uncertainty. We also propose an ensemble-aware loss to encourage the diversity between ensemble members and investigate the benefit of incorporating adversarial training in the video compression task. Experimental results on 1080p sequences show that our model can effectively save bits by more than 20% compared to DVC Pro (Lu et al., 2020b).

## 1    INTRODUCTION

Video contents are reported to account for 82% percent of all consumer Internet traffic by 2021 and are growing rapidly with an increasing demand for high-resolution videos (e.g., 4K movies) and live streaming services (Cisco, 2020). It is therefore critical for us to improve the video compression performance to transmit video with higher quality given limited Internet bandwidth. In recent years, there has been a surge of deep learning-based video compression models (Rippel et al., 2019; Lu et al., 2020b; Agustsson et al., 2020) and some of them have achieved comparable or even better performance than previous traditional video codecs, such as x.264 (Tomar, 2006).

Despite previous deep learning-based video codecs have achieved improved performance on many challenging datasets, most of the state-of-the-art models estimate deterministic predictions for intermediate representations, such as optical flows and residuals. These models fail to represent the aleatoric uncertainty inherent in the model inputs or the epistemic uncertainty in the model parameters and would blindly assume the predictions to be accurate, which is not always the case (Der Kiureghian & Ditlevsen, 2009; Kendall & Gal, 2017). In terms of video compression, such models produce deterministic motion vectors (or optical flows) and residuals for each pixel location, ignoring the fact that optical flows may not be estimated accurately in occluded regions and around object boundaries, and the quantization operation for lossless entropy coding also introduces additional noises to the inputs of the decoders. Underlying errors in such overconfident intermediate predictions are propagated to later stages of the P-frame model and even to subsequent frames for models built on temporal correlation, leading to suboptimal performance of the compression system.

Predictive uncertainty is crucial for us to understand how certain the model is about the predictions, especially for out-of-distribution data. However, most neural networks do not offer such information and tend to produce overconfident predictions (Gal, 2016; Lakshminarayanan et al., 2017). Bayesian neural networks (MacKay, 1992; Hinton & Neal, 1995) are widely used to quantify predictive uncertainty, but lack practicality due to greatly increased computation complexity and do not scale well to high dimensional data. Gal & Ghahramani (2016) proposed Monte Carlo dropout that performs test-time dropout. It is simple to implement but is not suitable for deep learning-based compression since it requires multiple decoding-time inferences and yields non-deterministic outputs.

In terms of deep learning-based video compression, two non-Bayesian approaches are considered to represent the predictive uncertainty: (1) modeling the uncertainty explicitly by regressing the empirical uncertainty of the model outputs (Nix & Weigend, 1994); and (2) using ensembles for predictive

uncertainty estimation (Lakshminarayanan et al., 2017). Agustsson et al. (2020) took the first approach and proposed scale-space flow for motion compensation. Specifically, their model predicts a scale field besides the standard 2-dimension flow field, representing the uncertainty associated with each MV. Gaussian blurring is then applied to the reference frame and the scale parameter is used to control the scale of the Gaussian kernel. Experimental results show that the self-supervised scale parameter increases around the boundaries of the objects and in regions where the motion is large, which confirms that the scale field learns to estimate the uncertainty of the optical flow prediction.

In this work, we consider the second approach and represent the underlying uncertainty with an ensemble of outputs. Ensembles of models have been shown to improve predictive performance (Dietterich, 2000), and ensembles of neural networks have also been used to boost model performance on many challenging benchmark datasets (Szegedy et al., 2015). Instead of producing a deterministic prediction, ensemble methods perform model combination and reflect the uncertainty of out-of-distribution data. Here we propose an ensemble-based multi-head decoding module that can generate an ensemble of intermediate outputs, such as motion vectors and residuals, and implicitly represent the predictive uncertainty with the variance of the Gaussian mixture prediction. We no longer assume that any individual intermediate prediction is accurate and explicitly consider multiple candidates of them. Such uncertainty is then propagated to later stages and all modules in our framework are optimized in an end-to-end fashion.

To further improve the performance of our ensemble-based video compression model, we propose an ensemble-aware loss to encourage diversity between different branches and incorporate an adversarial training strategy, fast gradient sign method (FGSM) (Goodfellow et al., 2014), to smooth the intermediate latent representations. Our experiments show that our model achieves an average bitrate saving of 15% when the distortion is measured in PSNR.

The contributions of this work can be summarized as follows:

- We are the first to leverage deep ensembles in a video compression model and demonstrate that our approach can be used to capture the underlying uncertainty of intermediate representations and prevent the model from being overconfident with intermediate predictions.
- We demonstrate that our approach can effectively improve the performance of deep-learning based video compression and can be widely applied to optical flow-based video codecs with negligible complexity increase.
- Our experiments show that our model outperforms previous state-of-the-art models such as DVC Pro (Lu et al., 2020b) and Lu et al. (2020a), and our ablation study proves the effectiveness of each module.

## 2 RELATED WORK

**Learned video compression.** Previous learning-based video compression methods can be categorized into two groups: (i) one-stage models, such as methods based on 3D autoencoders (Pessoa et al., 2020; Habibian et al., 2019); and (ii) two-stage models, which is adopted by most previous state-of-the-art methods, consist of predicted frame generation and residual coding. Lu et al. (2019) proposed an end-to-end trainable video codec, DVC, that utilizes an optical-flow network (Ranjan & Black, 2017) for motion compensation and then compresses the residuals. An extension of DVC, known as DVC Pro (Lu et al., 2020b), improves the compression performance by introducing refinement modules and auto-regressive entropy models. Feng et al. (2020) considered residual coding in the feature domain and choose between pixel-level residuals and feature-level residuals with RDO at encoding time. Agustsson et al. (2020) proposed scale-space flow to blur the intermediate reconstructions when motion vectors are not estimated well. Chen et al. (2021) proposed to represent videos with neural networks and achieved promising results for video compression.

**Model uncertainty.** Predictive uncertainty can be grouped into aleatoric uncertainty and epistemic uncertainty (Der Kiureghian & Ditlevsen, 2009). Aleatoric uncertainty captures the noises inherent in the observations and cannot be explained away with more data, while epistemic uncertainty accounts for uncertainty in the model structure or parameters and can be reduced with more training data. Bayesian neural networks (MacKay, 1992; Hinton & Neal, 1995) is a widely used approach for modeling predictive uncertainty that extends the traditional neural networks by learning a posterior distribution of model parameters from the observed data. Various non-Bayesian approaches

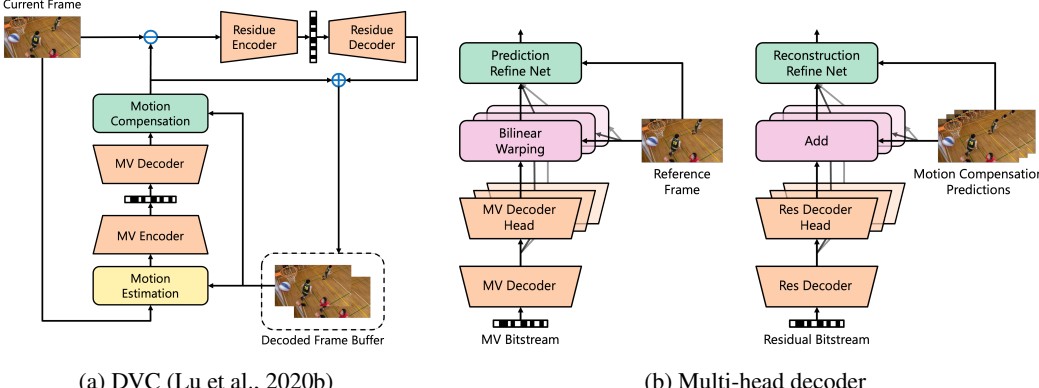

(a) DVC (Lu et al., 2020b)  (b) Multi-head decoder

Figure 1: (a) A low latency optical flow-based video compression, DVC (Lu et al., 2020b). (b) Our proposed multi-head decoder structure (Section 3.2). 'MV' stands for motion vectors, and 'Res' stands for residuals.

have also been proposed, such as utilizing the probabilities of softmax distributions (Hendrycks & Gimpel, 2016) and Specialists+1 Ensemble for representing the predictive uncertainty for adversarial samples (Abbasi & Gagné, 2017). Gal & Ghahramani (2016) proposed Monte Carlo dropout by performing multiple inferences with Dropout at test time. Lakshminarayanan et al. (2017) proposed to use an ensemble of neural networks for quantifying predictive uncertainty.

**Deep Ensembles.** The neural networks community has been investigating ensembles of deep networks since the early 1990s (Hansen & Salamon, 1990; Wolpert, 1992; Perrone & Cooper, 1993). Krogh & Vedelsby (1995) proved the bias-variance trade-off for ensemble models which suggested the importance of the diversity among ensemble members. Lee et al. (2015) investigated several training strategies to train an ensemble and proposed ensemble-aware oracle loss to encourage diversity. GoogLeNet (Szegedy et al., 2015), one of the best-performing models on ILSVRC 2014, is an ensembles of CNNs. Lakshminarayanan et al. (2017) proposed to estimate predictive uncertainty by training multiple stand-alone neural networks. Garipov et al. (2018); Fort et al. (2020) showed that deep ensembles can learn different modes of function with ensemble members that only differ in initialization weights.

## 3  UNCERTAINTY-AWARE DEEP VIDEO COMPRESSION

This section presents our main contributions. We introduce the theoretical background and the motivation of our proposed approach in Section 3.1. Then we introduce the ensemble-based multi-head structure to decode multiple candidates of motion vectors and residuals in Section 3.2. In order to encourage the diversity among the ensemble members and to improve the overall performance, we propose an ensemble-aware loss for multi-head decoders in Section 3.3. Finally, we introduce an adversarial training strategy that we find beneficial for the learning-based video compression task in Section 3.4.

### 3.1  UNCERTAINTIES IN DEEP VIDEO COMPRESSION

We consider the predictive coding-based framework, DVC, proposed by Lu et al. (2020b) (Fig. 1a). Let the current frame be $x_i$ and the reconstructed previous frame from the buffer be $\hat{x}_{i-1}$. We estimate a motion vector (MV) map $f_i$ with a motion estimation network. The optical flow is then sent to a motion auto-encoder for transform coding, yielding quantized bits $\hat{a}_i$ and the reconstructed optical flow $\hat{f}_i$. Bilinear warping is used for motion compensation (MC) and a MC prediction $\tilde{x}_i$ with residual $r_i = x_i - \tilde{x}_i$ is obtained. The residual $r_i$ is then compressed with a residual encoder and decoder, outputting quantized residual bit stream $\hat{b}_i$ and the decoded residual $\hat{r}_i$. The reconstructed current frame is the sum of the MC prediction and the decoded residual written as

$$\hat{x}_i = \text{BilinearWarp}(\hat{x}_{i-1}, \hat{f}_i) + \hat{r}_i. \tag{1}$$

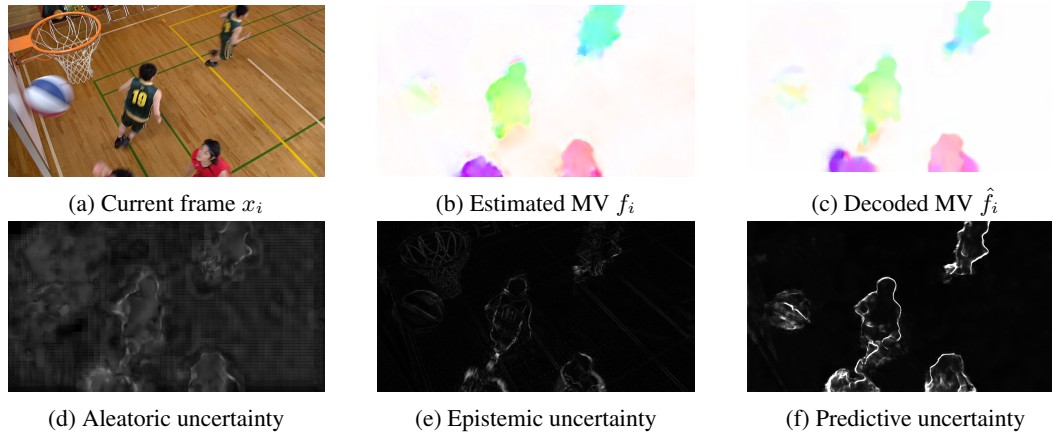

(a) Current frame $x_i$     (b) Estimated MV $f_i$     (c) Decoded MV $\hat{f}_i$

(d) Aleatoric uncertainty     (e) Epistemic uncertainty     (f) Predictive uncertainty

Figure 2: A preliminary experiment on the underlying uncertainty of the optical flows. (a) The current frame $x_i$ to be compressed. (b) The estimated MV $f_i$. (c) The decoded MV $\hat{f}_i$. (d) Aleatoric uncertainty measured as the L2 distance between two optical flows with and without a small perturbation on the bit stream. (e) Epistemic uncertainty measured by motion vectors that cannot be estimated well. (f) The predictive uncertainty represented by the multi-head decoder. Visualization of predictive uncertainty on other sequences can be found in Appendix A.6.

**Aleatoric uncertainty.** Although at encoding time we have full information necessary to decode $\hat{F}_i$, for lossy compression at certain bit rates, we quantize the bit stream that is passed to the decoder, and inevitably introduce aleatoric uncertainty at decoding time. Since the aleatoric uncertainty cannot be reduced with more training data, a well-trained codec cannot reduce the quantization noise or fully recover the estimated MV $f_i$.

Consider the MV auto-encoder in the DVC framework above. The lossy compression of the motion vectors can be summarized as

$$a_i = \text{MVEncoder}(f_i)$$
$$\hat{a}_i = q(a_i) = a_i + \eta$$
$$\hat{f}_i = \text{MVDecoder}(\hat{a}_i) \tag{2}$$

If the MV decoder is implemented with a linear model parameterized by $w$, the impact of the quantization noise $\eta$ on the decoded MV is given by

$$w^\top \hat{a}_i = w^\top (a_i + \eta) = w^\top a_i + w^\top \eta \tag{3}$$

Since $\eta$ is introduced by the quantization operation, we have $\|\eta\|_\infty \leq 1/2 = \varepsilon$ and it follows that the upper bound of the effects from the quantization operation is given by

$$\|w^\top \eta\|_1 \leq \varepsilon \left\| w^\top \text{sign}\left( \frac{\partial w^\top a_i}{\partial a_i} \right) \right\|_1 = \frac{1}{2} \|w^\top \text{sign}(w)\|_1 \tag{4}$$

While in practice the MV decoder is usually implemented with a stack of convolution layers and nonlinear activation layers, such as leaky ReLUs, the transformation of the MV decoder may be too linear to reject the quantization noise (Goodfellow et al., 2014).

We visualize the aleatoric uncertainty by adding a small perturbation $\|\eta_0\| \approx 0.2 \cdot \|\eta\|_\infty$ to the bit stream (see Appendix A.2) and the impact on the decoded MVs is depicted in Fig. 2d. It shows that there is more aleatoric uncertainty in regions where the motion is complicated and the impact due to the aleatoric uncertainty is nontrivial.

**Epistemic uncertainty.** Due to limited observed data during training, epistemic uncertainty accounts for the uncertainty in the estimated motion vectors. Motion vectors near the object boundaries and occluded regions tend not to be estimated well, and warping erroneous motion vectors would propagate errors to the residual coding. We may roughly visualize such uncertainty by optimizing

an motion estimation network with regards to the mean squared errors (MSE) between the current frame $x_i$ and the warped frame

$$\mathcal{L} = \text{MSE}(x_i, \text{BilinearWarp}(\hat{x}_{i-1}, f_i)) \tag{5}$$

We depict the results in Fig. 2e. Since the motion estimation is optimized to minimize the MSE, regions where the MSE is large are likely to have a larger epistemic uncertainty and the corresponding motion vectors cannot be estimated well given the limited training data.

Ideally, we could save MV bits by not encoding MVs that are not estimated well and save residual bits by not warping MVs that would not help to reduce residuals. This is often difficult to implement as an end-to-end trained deep neural network. In the next section, we will show that with the help of multi-head decoders, the model could learn to exploit available information in the bit stream and handle those predictions with larger uncertainty.

## 3.2 MULTI-HEAD DECODER

Our proposed multi-head decoder structure decodes multiple groups of motion vectors (MV) for motion compensation and multiple groups of residuals for final reconstruction. The multi-head MV decoder and multi-head residual decoder are depicted in Fig. 1b. Take the multi-head MV decoder as an example. The MV decoder base first decodes MV feature from the quantized MV bitstream $\hat{a}_i$. Then $h_{mv}$ groups of MVs, denoted by $\{\hat{f}_i^k \mid k = 1, \ldots, h_{mv}\}$, are decoded from the MV feature with respective MV decoder heads. We obtain $h_{mv}$ warped frames $\{\tilde{x}_i^k \mid k = 1, \ldots, h_{mv}\}$ by bilinearly warping each $\hat{f}_i^k$ on the reference frame $\hat{x}_{i-1}$. The $h_{mv}$ warped frames are then concatenated for motion compensation and retained for final reconstruction.

Many previous ensemble-based models train an ensemble of stand-alone neural networks (Szegedy et al., 2015; Abbasi & Gagné, 2017; Wang et al., 2020). While they can outperform the single-model baseline by a wide margin, the number of parameters are greatly increased, as well as the inference complexity. Lee et al. (2015) proposed to share backbone parameters with TreeNets, but the models achieve the best performance when very few layers are shared. In our multi-head decoder structure, each decoder branch shares most of the convolution layers, making each decoder head light weight. This design effectively improves the overall performance with negligible complexity increase (see Section 4.4).

The ensemble of decoded MVs can be represented by an equally weighted Gaussian mixture model given by

$$\hat{f}_i \sim \frac{1}{h_{mv}} \sum_{k=1}^{h_{mv}} \mathcal{N}(f \mid \hat{f}_i^k, \Sigma_i^k), \ \Sigma_i^k = \begin{bmatrix} \sigma_{i,x}^k & 0 \\ 0 & \sigma_{i,y}^k \end{bmatrix} \tag{6}$$

where $\sigma_{i,x}^k$ and $\sigma_{i,y}^k$ are the variance in $x$ and $y$ directions respectively. The mean and variance of the Gaussian mixture model are respectively (see Appendix A.3)

$$\mathbb{E}[\hat{f}_i] = \mu_{\hat{f}_i} = \frac{1}{h_{mv}} \sum_{k=1}^{h_{mv}} \hat{f}_i^k; \ \sigma_{\hat{f}_{i,x}}^2 = \mathbb{E}[\hat{f}_{i,x}^2] - \mu_{\hat{f}_i}^2 = \frac{1}{h_{mv}} \sum_{k=1}^{h_{mv}} \left( \left( \sigma_{i,x}^k \right)^2 + \left( \hat{f}_{i,x}^k \right)^2 \right) - \mu_{\hat{f}_i}^2 \tag{7}$$

Empirically, we can visualize the predictive uncertainty represented by this ensemble model with the variance of the Gaussian mixture model by setting $(\sigma_{i,x}^k)^2 = (\sigma_{i,y}^k)^2 = 1$, which gives

$$\sigma_{\hat{f}_{i,x}}^2 = \frac{1}{h_{mv}} \sum_{k=1}^{h_{mv}} \left( \hat{f}_{i,x}^k \right)^2 - \left( \frac{1}{h_{mv}} \sum_{k=1}^{h_{mv}} \hat{f}_i^k \right)^2 + 1 \tag{8}$$

The predictive uncertainty for the first two frames in the BasketballDrill sequence is depicted in Fig. 2e. We can see that the predictive uncertainty properly capture both the aleatoric uncertainty and the epistemic uncertainty shown in Fig. 2: the basketball has large aleatoric uncertainty due to rapid motion and the object boundaries have large epistemic uncertainty.

**Relation to scale-space flow.** Agustsson et al. (2020) estimated a scale field $g_z$ besides the 2-dimensional optical flow $(g_x, g_y)$. We may represent the decoded MV with a multivariate Gaussian

distribution given by

$$g_{mv} \sim \mathcal{N}((g_x, g_y)^\top, \Sigma), \ \Sigma = \begin{bmatrix} g_z & 0 \\ 0 & g_z \end{bmatrix} \tag{9}$$

and the scale-space warp gives a weighted mean of the warped value obtained from $g_{mv}$. The Gaussian mixture prediction from our proposed multi-head decoder can represent a more diverse predictive uncertainty distribution than the single Gaussian distribution in the scale-space flow. It has been shown in Garipov et al. (2018); Fort et al. (2019) that ensemble models produce diverse results by learning different modes of the function, rather than interpolating around a given mean in the output space. It should be noted that while we only regress the mean of MV predictions in each ensemble branch and model the predictive uncertainty with the variance of the Gaussian mixture prediction, the multi-head decoder can easily be extended to an ensemble of scale-space flows.

### 3.3 ENSEMBLE-AWARE TRAINING

Intuitively, diversity is a key factor for ensemble models. Ensemble members similar in the parameter space are unlikely to provide any more useful information than their single-model counterpart. Krogh & Vedelsby (1995) proved the bias-variance trade-off in ensemble, $E = \bar{E} - \bar{A}$, which suggested that the inherent variance is the key for the ensemble models to be effective and we should encourage the diversity among the ensemble members.

In the previous literature, multiple approaches are considered, including random initialization, bagging, and boosting. Randomly initializing the model parameters is a simple but effective approach to induce randomness and is quite suitable for deep ensembles (Lee et al., 2015). Bagging trains ensemble members on independently drawn examples with bootstrap sampling. Lee et al. (2015) showed that bagging may harm the model performance since each model may see only 63% of the available data and would perform poorly when there is high correlation inherent in the data (Bartlett et al., 1998). Boosting generates the ensemble models sequentially and can be quite time-consuming in terms of neural networks.

For deep compression models with our proposed multi-head decoder structure, there is a single model with multiple parallel ensemble branches. Bagging or boosting would not work for such network design since the model is trained on randomly sampled data but different ensemble branches would see the same data. Therefore, we choose to randomly initialize the network parameters and initial experiments show the effectiveness. To further encourage the diversity among the ensemble members, we propose an ensemble-aware loss for deep ensembles to induce additional randomness.

Given $h$ decoded outputs from the multi-head decoder, we obtain $h$ predictions (motion-compensated predictions or final reconstructions) $\bar{x}^t$ for $t = 1, \ldots, h$. The ensemble-aware loss is given by

$$\mathcal{L}_{\text{ensemble-aware}}(x, \bar{x}^1, \ldots, \bar{x}^h) = \sum_{t=1}^{h} \frac{1}{H \times W} \sum_{1 \le i \le H} \sum_{1 \le j \le W} \min_t \|\bar{x}_{i,j}^t - x_{i,j}\|_2^2 \tag{10}$$

and the minimum can be relaxed to choosing the smallest $k$ out of the $h$ norms. Our ensemble-aware loss encourage diversity between heads by relaxing the magnitude of the loss, and at the same time ensure performance of individual heads by updating every head at every iteration. Since the gradients propagated to each head depend on the head with the minimal MSE, rather than the MSE of the head itself, we can effectively induce randomness and therefore encourage diversity. Although the oracle set loss proposed in Lee et al. (2015) helps to encourage diversity and boost the oracle accuracy, it would significantly harm the performance of individual ensemble members since each head only sees a small portion of all training data. Experiments on learned video compression also showed that adopting the oracle set loss will decrease the overall performance. Instead, our ensemble-aware loss can effectively encourage diversity among ensemble members, and at the same time, each individual member sees all training data and achieves reliable performance.

### 3.4 ADVERSARIAL TRAINING WITH FGSM

Adversarial examples (Szegedy et al., 2013) are training samples with a small but non-random perturbation that are misclassified by neural networks with high confidence. Goodfellow et al. (2014)

proposed the fast gradient sign method (FGSM) that applies linear but intentionally worst-case perturbation to the training samples, which is given by

$$\eta = \epsilon \cdot \mathrm{sign}(\nabla x J(\theta, x, y)) \tag{11}$$

where $J(\theta, x, y)$ is the cost function, and $\epsilon$ controls the norm of the perturbation. This adversarial training strategy has been shown to boost the image classification performance, as well as improve the model's robustness to adversarial examples. Lakshminarayanan et al. (2017) interpreted FGSM as an efficient solution to smooth the predictive distributions by increasing the likelihood of the target around an $\epsilon$-neighborhood of the observed training samples.

We find adversarial training with FGSM closely related to learned lossy compression and an effective approach to improve the performance of learned video codecs. In transform coding, we want the latent representation to be as smooth as possible, since after quantization, all latent representations in the $\epsilon$-neighborhood, $\{\hat{a} + \eta \mid \|\eta\|_\infty < \epsilon\}$, corresponds to the same decoded output. Learning a smooth latent representation would help to make the outputs more robust to quantization noise. Although this could be a natural result of an end-to-end optimized video codec, the experimental results show that FGSM can effectively improve the rate-distortion performance.

## 4 EXPERIMENTS

### 4.1 EXPERIMENTAL SETUP

**Model architecture.** Our base model architecture follows the design in Lu et al. (2020b) and we use auto-regressive and hierarchical priors for both the motion vector and residual compression. In order to optimize the model in an end-to-end manner, we need to relax the bits estimation since quantizing the latent bits would make the gradients zero almost everywhere. Following Ballé et al. (2016), we substitute the quantization operation with additive uniform noise during training and perform actual quantization during inference. The structure of multi-head decoder modules are depicted in Figure 1b and more technical details on the implementation are available in Appendix A.1.

**Training datasets.** Our model is trained on 64,612 video sequences from the training part in Vimeo-90K settuplet dataset (Xue et al., 2019). Each video clip has 7 frames with a resolution of $448 \times 256$. During training, we randomly crop the video sequences into $256 \times 256$ pixels. Given two successive frames from a random sequence, we treat the first frame as the reference frame and our model is trained to minimize the rate-distortion cost of encoding and decoding the second frame.

**Training details.** In our experiments, we adopt the progressive training strategy and warm up the inter-coding module for 150,000 steps with the ensemble-aware motion compensation loss in Equation 10. Then the model is end-to-end optimized with the rate-distortion loss given by

$$\mathcal{L}_{RD} = (R_w(\hat{w}_i) + R_z(\hat{z}_i)) + \lambda \cdot D(x_i, \hat{x}_i) \tag{12}$$

where $R_w(\hat{w}_i)$ and $R_z(\hat{z}_i)$ represent the numbers of bits used to encode the motion vectors and the residual, $D(F_i, \hat{F}_i)$ measures the distortion in mean squared error MSE or multi-scale structural similarity MS-SSIM (Wang et al., 2003), and $\lambda$ is the hyperparameter controlling the trade-off. Four models are trained with different quality rates by setting $\lambda = 256, 512, 1024, 2048$. We use the AdamW optimizer (Loshchilov & Hutter, 2017) with an initial learning rate of $1 \times 10^{-4}$, which is then decreased to $1 \times 10^{-5}$. Each model is trained on one NVIDIA V100 GPU.

### 4.2 QUANTITATIVE RESULTS

To show the effectiveness of our proposed uncertainty-aware model, we test our model on the first 100 frames from video sequences in HEVC (Sullivan et al., 2012) with GoP size 10, and the first 120 frames from sequences in UVG (Mercat et al., 2020), MCL-JCV (Wang et al., 2016) with GoP size 12. To balance the trade-off between complexity and performance, we choose $h_{mv} = h_{res} = 4$. In order to build the best learning-based video codec, we adopt the state-of-the-art image compression model (Cheng et al., 2020) for intra frame coding. In order to fairly compare the performance, we test the DVC and DVC Pro with the same same intra frame model, denoted as DVC (cheng2020) and DVC Pro (cheng2020). For some of the previous state-of-the-art deep video compression methods that are not available, such as HU_ECCV20 (Hu et al., 2020) and LU_ECCV20 (Lu et al., 2020a),

Table 1: Quantitative comparisons between different learning-based video compression models measured in BD rate. The anchor model is x265 (*veryslow*). Negative number means bitrate saving and positive number means bitrate increase.

| MODEL | HEVC B | HEVC C | HEVC D | HEVC E | UVG | MCL-JCV |
|---|---|---|---|---|---|---|
| x264 (*veryslow*) | 35.0% | 19.9% | 15.5% | 50.0% | 32.7% | 30.3% |
| x265 (*veryslow*) | 0.0% | 0.0% | 0.0% | 0.0% | 0.0% | 0.0% |
| DVC (public) | 26.7% | 41.5% | 31.1% | 17.8% | 21.9% | 16.0% |
| DVC Pro (public) | -0.4% | 11.5% | 4.5% | -3.8% | -12.6% | -5.3% |
| DVC (cheng2020) | 7.9% | 15.1% | 7.2% | 21.1% | 17.2% | 13.3% |
| DVC Pro (cheng2020) | -9.0% | 7.2% | -6.9% | 17.2% | -7.9% | -4.1% |
| HU_ECCV20 | 2.4% | 13.0% | 10.8% | -8.6% | -5.4% | -12.6% |
| LU_ECCV20 | 5.0% | 8.4% | 3.6% | 11.7% | 8.8% | 8.4% |
| **Ours** | **-22.3%** | **-6.0%** | **-19.0%** | **-24.3%** | **-25.5%** | **-18.2%** |

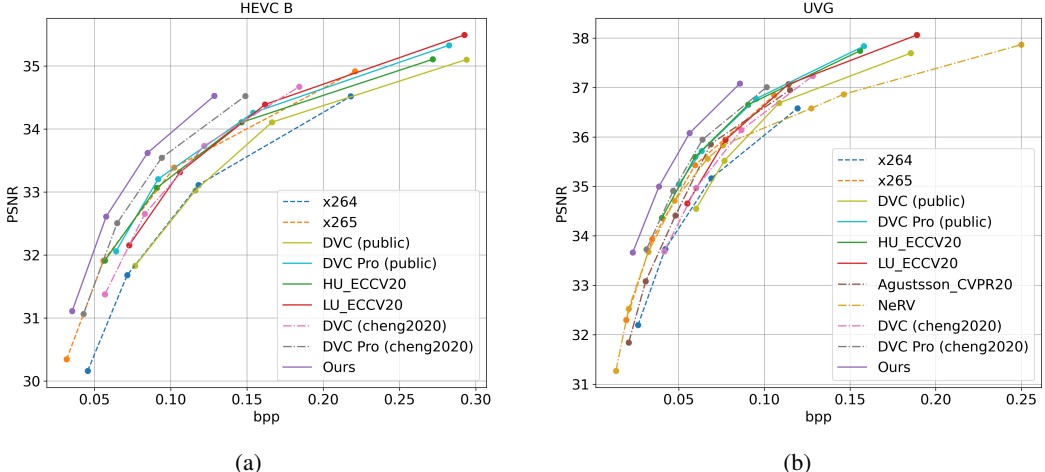

(a)          (b)

Figure 3: Rate-distortion comparison between our model and x264 (*veryslow*), x265 (*veryslow*), Hu_ECCV20 (Hu et al., 2020), LU_ECCV20 (Lu et al., 2020a), Agustsson_CVPR20 (Agustsson et al., 2020), and NeRV (Chen et al., 2021) on sequences from HEVC Class B and UVG. Results on other HEVC sequences and MCL-JCV are reported in Appendix A.4. Best viewed in color.

we use the public results reported in Hu (2020). We calculate the BD-rate (Bjøntegaard, 2001) of different learning-based video compression models using x.265 as the anchor model and the results on HEVC, UVG, MCL-JCV are reported in Table 1. We also plot the RD curves of different codec models in Fig. 3. More RD curve plots are available in Appendix A.4.

From the results reported in Table 1 and Fig. 3, we could see that our proposed model can effectively save bits compared to our strong baseline and outperform previous state-of-the-art learning-based video codecs by a wide margin in all testing datasets.

## 4.3 QUALITATIVE RESULTS

In Fig. 2 we visualize the aleatoric uncertainty and epistemic uncertainty in the first two frames of the BasketballDrill sequence, as well as the predictive uncertainty represented by the multi-head MV decoder. As we can see, the predictive uncertainty is larger in regions where the motion cannot be estimated well or too complicated to encode. Our uncertainty-aware model learns to effectively represent the underlying uncertainty with an ensemble of decoded MVs and such uncertainty is retained until the final reconstruction. More visualizations of the unsupervised predictive uncertainty can be found in Appendix A.6.

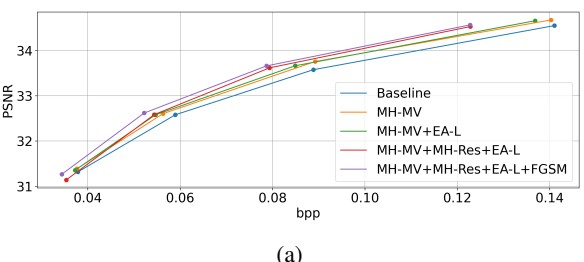 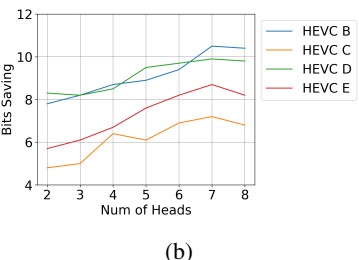

(a)                                                              (b)

Figure 4: (a) Effectiveness of various proposed module. (b) Ablation study on the number of heads in multi-head decoders.

## 4.4 ABLATION STUDY

**Effectiveness of various proposed modules.** We evaluate the effectiveness of multi-head decoders, ensemble-aware loss, and adversarial training with FGSM by running ablation experiments on the first 30 frames from all sequences in the HEVC dataset. We adopt the short training strategy for fast experimentation. The RD curves are presented in Fig. 4a and the BD-rates are reported in A.8. MH-MV and MH-Res stand for multi-head MV decoder and multi-head residual decoder, and EA-L refers to training with ensemble-aware loss. The results show the effectiveness of the multi-head decoder modules and the training strategies.

**Ablation study on the number of decoder heads.** We investigate the model's performance with different number of heads in the multi-head decoder on HEVC sequences. As shown in Fig. 4b, we train eight models with $h_{mv} = 1, 2, 3, 4, 5, 6, 7, 8$ where $h_{mv} = 1$ is the baseline without ensembling. We see that the multi-head decoder module is effective even with only two decoder heads and the performance is improved with more decoder heads. Quantitative results are reported in A.8.

**Complexity analysis.** Most previous deep ensembles train multiple stand-alone models (Szegedy et al., 2015; Abbasi & Gagné, 2017; Wang et al., 2020) or share very few shallow layers (Lee et al., 2015) for the model to be effective. With an ensemble of 6 models, the inference complexity (in MACs) and model size (in number of parameters) easily increase by $500\%$. With the help of multi-head decoders, our ensemble-based model share the backbone features and achieve superior results with limited complexity increase. For one extra MV and residual decoder added, the complexity increases by approximately $7\%$ and just $1\%$ in the model size. For the largest model we consider, where $h_{mv} = h_{res} = 8$, there is only a $48\%$ increase in complexity and $10\%$ in model size.

## 5 DISCUSSION

In this paper, we studied the aleatoric uncertainty and epistemic uncertainty in deep learning-based video compression and proposed to utilize an ensemble of intermediate predictions to represent the predictive uncertainty at decoding time. With multi-head decoders, our model can properly model the uncertainties in the decoded MVs or residuals, which would effectively avoid propagating underlying errors in a single deterministic prediction.

We investigated the performance of our uncertainty-aware decoding module and proposed a novel ensemble-aware loss to boost the diversity among the parallel ensemble branches in a single model. We also proposed to incorporate adversarial training for learning-based video codecs. Experimental results show the effectiveness of our approach.

Compared to one-stage learning-based video compression models, such as those based on 3D autoencoders (Pessoa et al., 2020; Habibian et al., 2019), two-stage motion compensation-based models can decode high-quality frames with low-latency. However, intermediate predictions in these two-stage pipelines are not always accurate and erroneous predictions could severely harm the performance of later stages, especially for out-of-distribution data. Therefore, it is critical to represent the predictive uncertainty and our proposed multi-head decoder is a simple but very effective approach to capture such uncertainty. Future directions could involve modules on the encoder side to model and propagate the uncertainty to the decoders for an end-to-end uncertainty-awareness.

## REPRODUCIBILITY STATEMENT

Key results in this paper are reproducible and in this section, we will refer to details helpful to reproduce the results. The implementation of our model are modified from the codes released by Bégaint et al. (2020) and Hu (2020). We propose the multi-head decoders in Section 3.2 and can be easily implemented by following the flow charts in Figure 1b and technical details in Appendix A.1. In Section 4.1 we introduce the dataset we use to train the model. The Vimeo-90K settuplet dataset (Xue et al., 2019) is publicly available and we fit our model on the training part of the dataset. Also in Section 4.1 we summarize our training strategy, which includes an ensemble-aware loss for model warm-up and then minimizing the rate-distortion cost until convergence. The motivation and formality of the ensemble-aware loss can be found in Section 3.3. This loss can be easily implemented in PyTorch. Since directly clamping the loss gives zero gradients almost everywhere, we create a custom autograd function to forward the gradients before and after the clamp operation and properly update the weights of each head.

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

## A  APPENDIX

### A.1  IMPLEMENTATION OF MULTI-HEAD DECODERS

As depicted in Figure 1b, multi-head decoders consist of a decoder backbone and multiple decoder heads. The decoder heads are light-weight and include two convolution layers with one leaky ReLU in between. For multi-head MV decoder, the decoder backbone first decodes MV feature from the MV bitstream $\hat{a}_i$, and then $h$ MVs are decoded with respective MV decoder heads. From the $h$ decoded MVs and the previous decoded frame, we obtain $h$ MC predictions with bilinear warping. The $h$ MC predictions are concatenated with the previous decoded frame and sent to the Prediction Refine Net, from which we get $h$ refined MC predictions. For multi-head residual decoder, the decoder backbone first decodes residual feature from the residual bitstream $\hat{b}_i$, and then $h$ residuals are decoded with respective residual decoder heads. From the $h$ decoded residuals and the $h$ refined MC predictions, we obtain $h$ reconstructions. The $h$ reconstructions are concatenated with $h$ refined MC predictions and sent to the Reconstruction Refine Net, from which we get 1 refined reconstruction as the final decoded frame. All modules in our model, including decoder backbone, decoder head, and refine nets, are implemented with neural networks and optimized in an end-to-end manner.

### A.2  VISUALIZING THE ALEATORIC UNCERTAINTY

To visualize the aleatoric uncertainty introduced from the quantization operation, we conduct preliminary experiments. We add a small perturbation $\eta_0$ to the quantized bit stream and obtain $\hat{w}_i' = \hat{w}_i + \eta_0$. The perturbation $\eta_0$ is only added to positions where the quantization gap is at least $0.1$ and corresponds to only $20\%$ of the size of the perturbation gap. We model the aleatoric uncertainty with the L1 norm between the two optical flows decoded from bit streams that differs in a small perturbation. Results on the first two frames of the BasketballDrill sequence are shown in Figure 2d. As we can see, the aleatoric uncertainty is not uniform across the whole image. Instead, there are more aleatoric uncertainty around the object boundaries and regions where the motion is large. While by definition, such aleatoric uncertainty cannot be reduced away, blindly assuming the optical flows to be accurate would lead to larger intermediate residual errors and cost more bits in the residual coding.

### A.3  MV PREDICTION AS A GAUSSIAN MIXTURE MODEL

The multi-head MV decoder predicts an ensemble of motion vectors $\{\hat{f}_i^k \mid k = 1, \ldots, h_{mv}\}$ from the MV bit stream and we can represent the prediction as an equally-weighted Gaussian mixture model given by

$$\hat{f}_i \sim \frac{1}{h_{mv}} \sum_{k=1}^{h_{mv}} \mathcal{N}(f \mid \hat{f}_i^k, \Sigma_i^k), \ \Sigma_i^k = \begin{bmatrix} \sigma_{i,x}^k & 0 \\ 0 & \sigma_{i,y}^k \end{bmatrix} \tag{13}$$

Since the Gaussian distributions are equally-weighted, the mean of the mixture distribution is

$$\mathbb{E}[\hat{f}_i] = \mu_{\hat{f}_i} = \frac{1}{h_{mv}} \sum_{k=1}^{h_{mv}} \hat{f}_i^k \tag{14}$$

and the variance of the mixture distribution in each direction is

$$
\begin{aligned}
\sigma^2_{\hat{f}_{i,x}} &= \mathbb{E}[\hat{f}^2_{i,x}] - \mu^2_{\hat{f}_i} \\
&= \frac{1}{h_{mv}} \sum_{k=1}^{h_{mv}} \mathbb{E}\left[\left(\hat{f}^k_{i,x}\right)^2\right] - \mu^2_{\hat{f}_i} \\
&= \frac{1}{h_{mv}} \sum_{k=1}^{h_{mv}} \left(\left(\sigma^k_{i,x}\right)^2 + \left(\hat{f}^k_{i,x}\right)^2\right) - \mu^2_{\hat{f}_i}
\end{aligned} \tag{15}
$$

In order to visualize the predictive uncertainty represented by this ensemble model, we set $(\sigma^k_{i,x})^2 = (\sigma^k_{i,y})^2 = 1$ and obtain

$$
\begin{aligned}
\sigma^2_{\hat{f}_{i,x}} &= \frac{1}{h_{mv}} \sum_{k=1}^{h_{mv}} \left(1 + \left(\hat{f}^k_{i,x}\right)^2\right) - \mu^2_{\hat{f}_i} \\
&= \frac{1}{h_{mv}} \sum_{k=1}^{h_{mv}} \left(\hat{f}^k_{i,x}\right)^2 - \left(\frac{1}{h_{mv}} \sum_{k=1}^{h_{mv}} \hat{f}^k_i\right)^2 + 1
\end{aligned} \tag{16}
$$

## A.4 QUANTITATIVE RESULTS

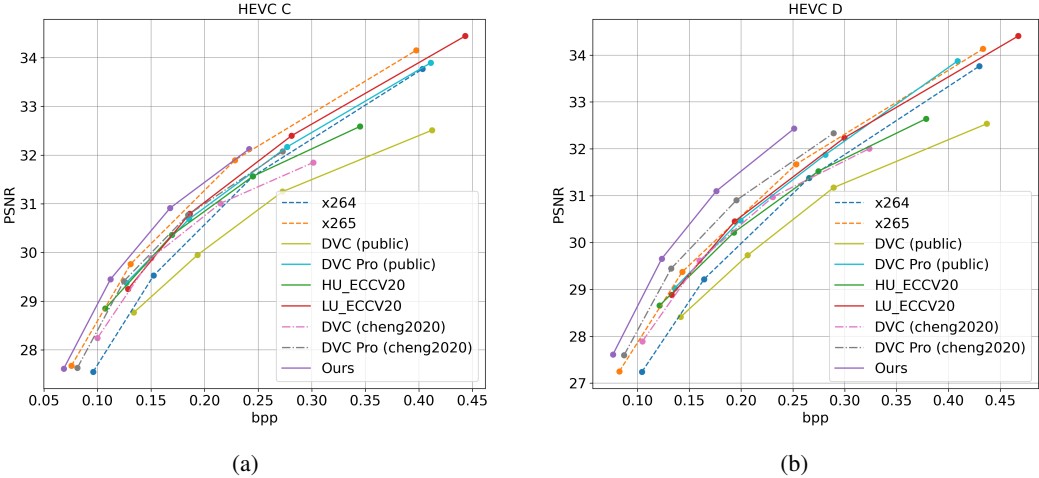

(a)  (b)

Figure 5: Rate-distortion comparison between our model and x264 (*veryslow*), x265 (*veryslow*), Hu_ECCV20 (Hu et al., 2020), LU_ECCV20 (Lu et al., 2020a), Agustsson_CVPR20 (Agustsson et al., 2020) on sequences from HEVC Class C and HEVC Class D. Best viewed in color.

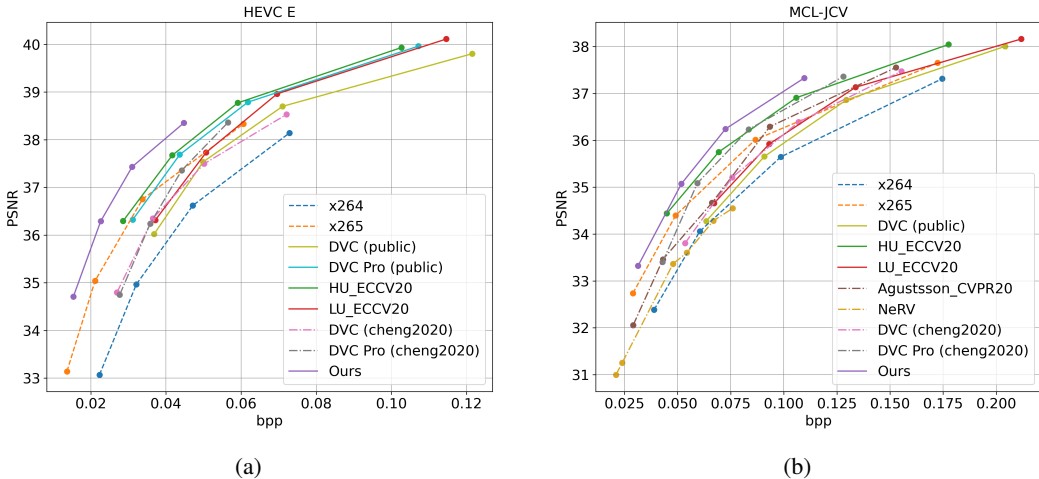

Figure 6: Rate-distortion comparison between our model and x264 (*veryslow*), x265 (*veryslow*), Hu_ECCV20 (Hu et al., 2020), LU_ECCV20 (Lu et al., 2020a), Agustsson_CVPR20 (Agustsson et al., 2020) on sequences from HEVC Class E and MCL-JCV. Best viewed in color.

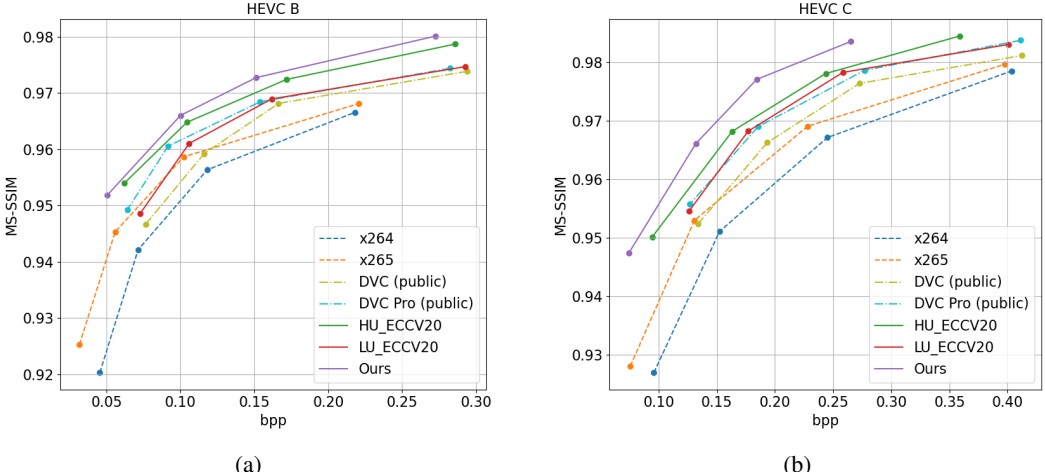

Figure 7: Rate-distortion comparison between our model and x264 (*veryslow*), x265 (*veryslow*), Hu_ECCV20 (Hu et al., 2020), LU_ECCV20 (Lu et al., 2020a), Agustsson_CVPR20 (Agustsson et al., 2020) on sequences from HEVC Class B and HEVC Class C. Distortion measured in MS-SSIM. Best viewed in color.

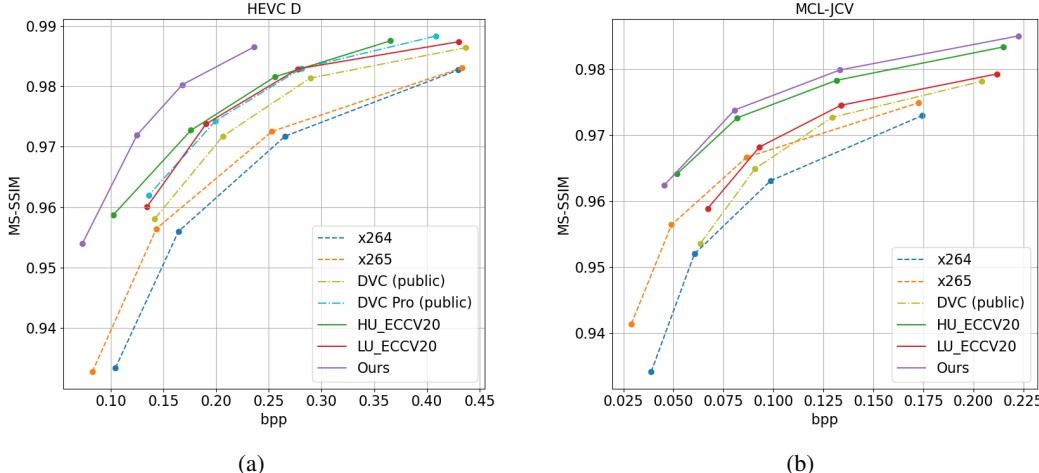

Figure 8: Rate-distortion comparison between our model and x264 (*veryslow*), x265 (*veryslow*), Hu_ECCV20 (Hu et al., 2020), LU_ECCV20 (Lu et al., 2020a), Agustsson_CVPR20 (Agustsson et al., 2020), and NeRV (Chen et al., 2021) on sequences from HEVC Class D and MCL-JCV. Distortion measured in MS-SSIM. Best viewed in color.

## A.5 MODEL PERFORMANCE USING DIFFERENT I-FRAME MODELS

In this section, we investigate the performance of our proposed model using different I-frame models on HEVC Class B and UVG. In Section 4.2, we report the experimental results of our proposed model using Cheng2020 (Cheng et al., 2020) as the I-frame model. Compared to DVC Pro, our proposed model with multi-head decoders can effectively reduce the bpp while maintaining the same reconstruction quality. As shown in Figure 3, this brings our model to a range of lower bitrates. In order to investigate our model performance in higher bitrates, we retrain our model with $\lambda = 4096$ and use VTM (VTM) as the I-frame model. The performance of our model and DVC Pro using VTM are reported in Figure 9. We could see that using VTM instead of cheng2020 as the I-frame model brings both our proposed model and DVC Pro to higher bitrates and similar conclusions can be made. Our proposed model can still outperform previous state-of-the-art models in higher bitrates and the multi-head decoders can bring significant bitrate savings regardless of the I-frame model used or the range of bitrates.

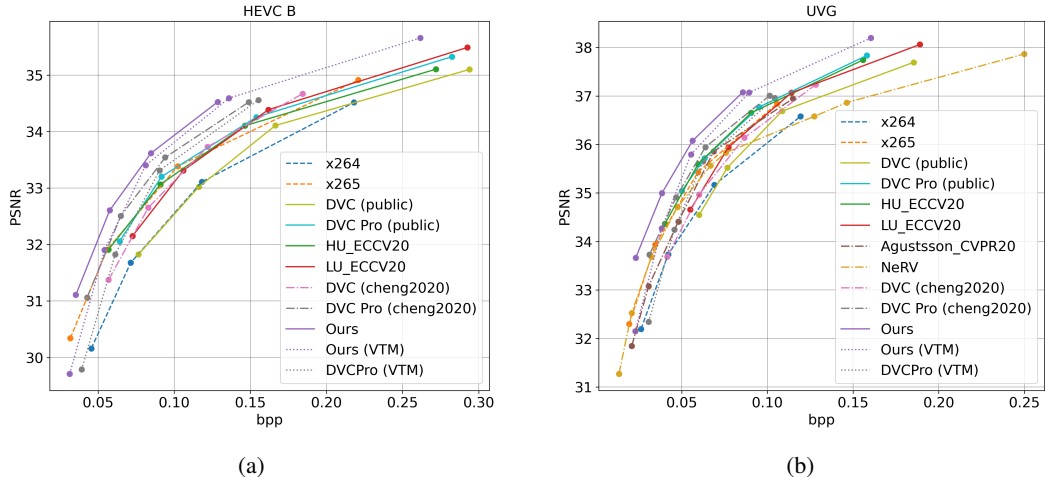

(a)            (b)

Figure 9: Rate-distortion comparison between our model and x264 (*veryslow*), x265 (*veryslow*), Hu_ECCV20 (Hu et al., 2020), LU_ECCV20 (Lu et al., 2020a), Agustsson_CVPR20 (Agustsson et al., 2020), and NeRV (Chen et al., 2021) on sequences from HEVC Class B and UVG. Distortion measured in PSNR. Best viewed in color.

## A.6 VISUALIZATION OF UNSUPERVISED PREDICTIVE UNCERTAINTY

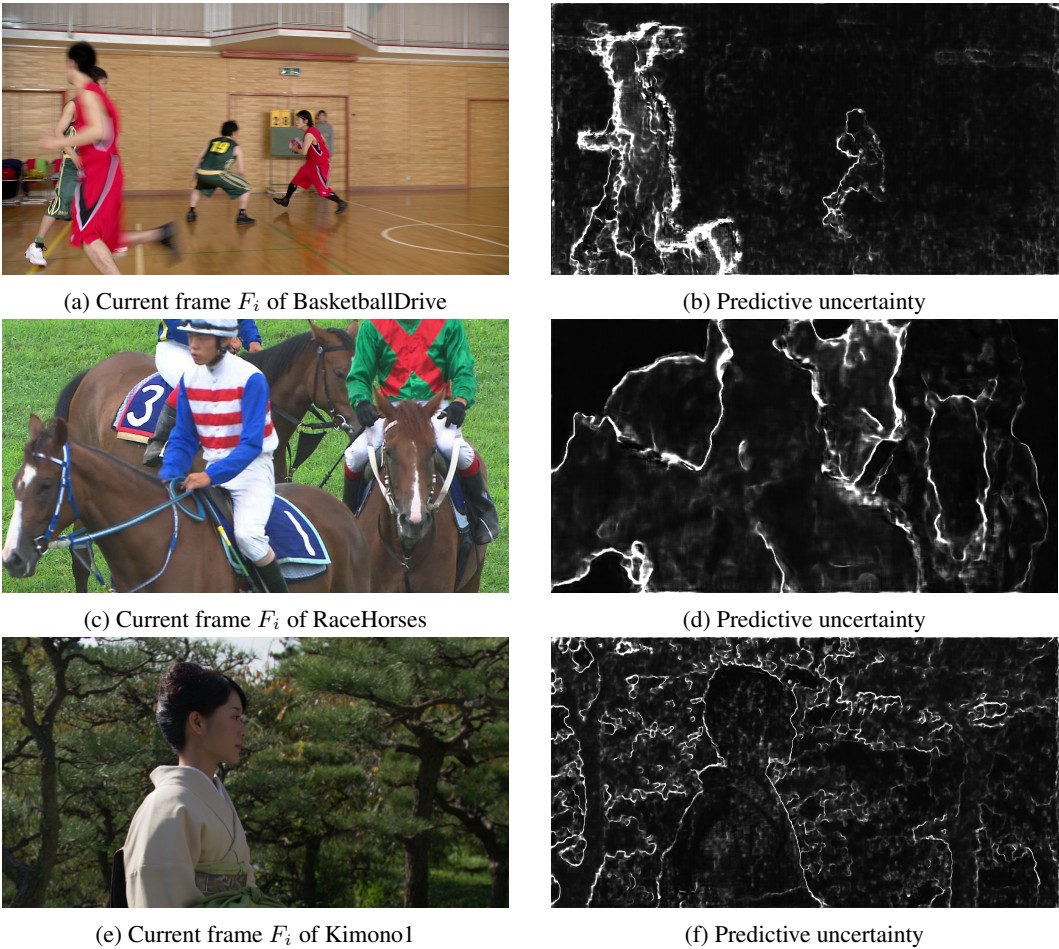

(a) Current frame $F_i$ of BasketballDrive

(b) Predictive uncertainty

(c) Current frame $F_i$ of RaceHorses

(d) Predictive uncertainty

(e) Current frame $F_i$ of Kimono1

(f) Predictive uncertainty

Figure 10: Visualization of the predictive uncertainty represented by our proposed multi-head decoder on BasketballDrive, RaceHorses, and Kimono1.

## A.7 QUALITATIVE RESULTS OF DECODED FRAMES

In this section, we show the qualitative examples of decoded frames for subjective comparison. We compare the performance of our proposed model with the baseline DVC Pro to show the strength and weakness of multi-head decoders. In sequence BQTerrace from HEVC B, our proposed model achieves a bitrate saving of 20.5% over DVC Pro. Sequence 20 from MCL-JCV is the only sequence where our model failed to outperform DVC Pro with a bitrate increase of 2.2%.

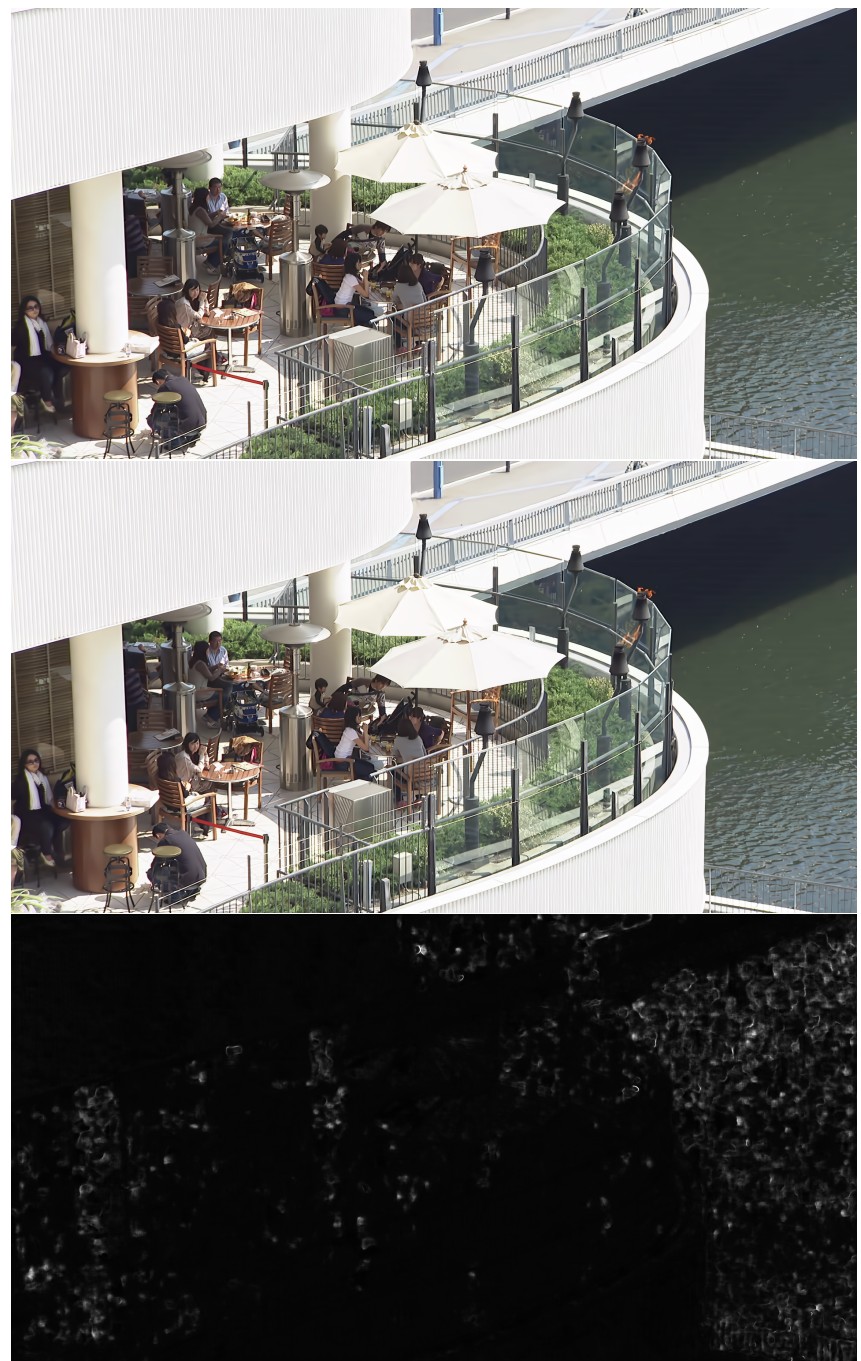

Figure 11: Comparison between our proposed model with DVC Pro on a decoded P-frame from sequence BQTerrace in HEVC B. In this sequence, our proposed model has a bitrate saving of 20.5% over DVC Pro. Top: our proposed model (0.01472 bpp); middle: DVC Pro (0.02587 bpp); bottom: predictive uncertainty.

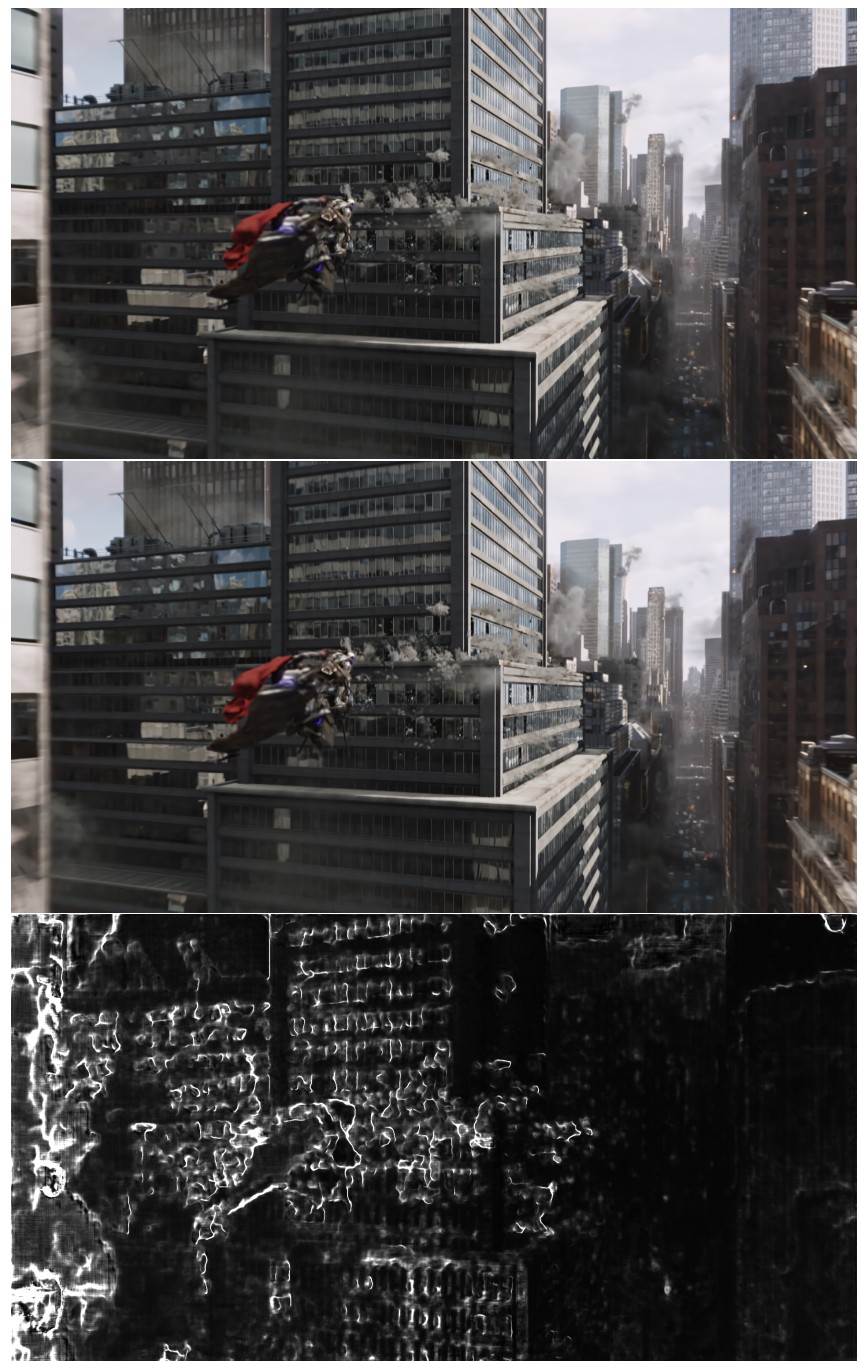

Figure 12: Comparison between our proposed model with DVC Pro on a decoded P-frame from sequence 20 in MCL-JCV. This is the only sequence where our proposed model failed to outperform DVC Pro with a bitrate increase of 2.2%. Top: our proposed model (0.10027 bpp); middle: DVC Pro (0.11786 bpp); bottom: predictive uncertainty.

## A.8 Detailed Ablation Study Results

To show the effectiveness of various proposed modules and to compare between models with different number of decoder heads, we train multiple models on the Vimeo-90K dataset with the fast training strategy and test the models on the first 30 frames in each HEVC sequence. We measure

the model performance with BD rates and the quantitative results are reported in Table 2 and Table 3 respectively.

Table 2: Ablation study on the effectiveness of each module. The performance are measured in BD rates using our baseline model as the anchor. **MH-MV**: multi-head MV decoder, **MH-Res**: multi-head residual decoder, **EA-L**: ensemble-aware loss, **FGSM**: adversarial training with FGSM. All models have $h_{mv} = h_{res} = 4$ considering the trade-off between performance and complexity.

| Setting | HEVC B | HEVC C | HEVC D | HEVC E |
|---|---|---|---|---|
| MH-MV | -5.8 | -3.1 | -4.0 | -3.7 |
| MH-MV + EA-L | -7.0 | -4.0 | -6.7 | -4.8 |
| MH-MV + MH-Res + EA-L | -8.7 | -6.4 | -8.5 | -6.7 |
| MH-MV + MH-Res + EA-L + FGSM | -12.7 | -7.9 | -11.2 | -11.4 |

Table 3: Ablation study on the number of heads in a multi-head decoder. The performance are measured in BD rates using our single-head baseline as the anchor. For simplicity, we set $h_{mv} = h_{res}$ and train eight models with $h_{mv} = h_{res} = 1, \ldots, 8$ using the fast training strategy.

| Setting | HEVC B | HEVC C | HEVC D | HEVC E |
|---|---|---|---|---|
| $h_{mv} = h_{res} = 1$ | 0.0 | 0.0 | 0.0 | 0.0 |
| $h_{mv} = h_{res} = 2$ | -7.8 | -4.8 | -8.3 | -5.7 |
| $h_{mv} = h_{res} = 3$ | -8.2 | -5.0 | -8.2 | -6.1 |
| $h_{mv} = h_{res} = 4$ | -8.7 | -6.4 | -8.5 | -6.7 |
| $h_{mv} = h_{res} = 5$ | -8.9 | -6.1 | -9.5 | -7.6 |
| $h_{mv} = h_{res} = 6$ | -9.4 | -6.9 | -9.7 | -8.2 |
| $h_{mv} = h_{res} = 7$ | -10.5 | -7.2 | -9.9 | -8.7 |
| $h_{mv} = h_{res} = 8$ | -10.4 | -6.8 | -9.8 | -8.2 |

