# OpenReview forum: "Uncertainty-Aware Deep Video Compression with Ensembles"
_ICLR.cc/2022/Conference — ICLR 2022 Submitted_

### Official Review · Reviewer_BZ5c · 2021-11-02

**Correctness:** 3
**Technical Novelty And Significance:** 2
**Empirical Novelty And Significance:** 3
**Recommendation:** 6
**Confidence:** 3

**Main Review:**

Here are issues or concerns regarding this paper.
(1) Both F_i and x_i are used to denote the current frame, e.g., in Fig. 2 (a) and in Sec. 3.1.
(2) What are the differences between the raw prediction and motion compensation prediction?
(3) Section 3.3 explains that the ensemble-aware loss can be relaxed to have the k-smallest outputs.
Can the relaxed prediction improve the prediction performance? If the k=1 is the best choice, why is that so?
(4) There is a broken sentence at the end of Sec. 4.1.
(5) What are the definitions for the Prediction Refine Net and Reconstruction Refine Net in Fig. 1(b)?
Are they neural networks that are trained in the framework?
If so, does the prediction refine net take multiple predictions and refine them simultaneously?
If not, I think the authors should clarify how the underlying uncertainties are predicted and utilized to better compress the video in the entire framework.
(6) Predictive uncertainties are analyzed by assuming the predictions from multi-head decoders follow the mixture of Gaussian model.
I think the number of decoders should be sufficiently large to support this claim.
(7) Check for typos, punctuations, and articles.
(8) Dependency on the optical flow estimation method.
(9) Failure cases?






**Summary Of The Paper:**

This paper presents an uncertainty-aware video compression framework with an ensemble of MV/residual decoders. To train the network, they use the ensemble-aware loss function which minimizes the loss function for the k best predictions. Moreover, this paper also analyzes underlying uncertainties in video compression, i.e., aleatoric and epistemic uncertainties, with visualization techniques for those uncertainties. Experimental results show that the proposed method can improve the compression performance by about 20% while not significantly increasing the computational complexity.

**Summary Of The Review:**

Overall, the paper was interesting because of the in-depth analysis and insights. However, I have a few concerns as stated in the main review. I would like to hear the answers to (3), (5), (6), (9) from the authors.

---

> ### Author Response · Authors · 2021-11-17
> **Response to Reviewer BZ5c - Part 1**
>
> We thank the reviewer for the feedback, and we address the concerns below:
>
> > Both F_i and x_i are used to denote the current frame, e.g., in Fig. 2 (a) and in Sec. 3.1.
>
> We thank the reviewer for pointing out the inconsistencies in notations. We apologize for the typos and have fixed these issues in the updated submission.
>
> > What are the differences between the raw prediction and motion compensation prediction?
>
> In Section 3.1, the raw prediction and motion compensation (MC) prediction refers to the same idea -- the predicted frame by exploiting the temporal dependencies with motion compensation. We revised the terms and wording in Section 3.1 to avoid such confusion.
>
> > Section 3.3 explains that the ensemble-aware loss can be relaxed to have the k-smallest outputs. Can the relaxed prediction improve the prediction performance? If the k=1 is the best choice, why is that so?
>
> It depends on the number of heads in the multi-head decoders. Our main quantitative results are based on a model with h=4, and k=1 and k=2 achieves the best performance. In our ablation study we also experimented with models with more heads and when h=8, choosing k=2 or k=4 can improve the prediction performance.
>
> In fact, k=1 and k=h are the two ends of a spectrum. **Ensemble-aware loss with k=1 allows the maximum diversity but cannot guarantee performance of individual heads; and ensemble-aware loss with k=h degenerates to common MSE, ensuring individual performance and allowing no diversity.** We would like to balance the trade-off between the two, and in our experiments, we use k=1 when there are four or fewer heads and increase k when there are more heads. We believe choosing the optimal value of k depends on specific tasks and models. Compared to ensemble-aware losses proposed in previous works, our proposed ensemble-aware loss has the flexibility to adapt to and increase the performance of models with different numbers of ensemble members. With the rise of deep ensembles and multi-head structures, we believe this will be an important contribution to the community.
>
> > There is a broken sentence at the end of Sec. 4.1.
>
> We thank the reviewer for pointing this out and we have fixed this error in the updated submission.
>
> > What are the definitions for the Prediction Refine Net and Reconstruction Refine Net in Fig. 1(b)? Are they neural networks that are trained in the framework? If so, does the prediction refine net take multiple predictions and refine them simultaneously? If not, I think the authors should clarify how the underlying uncertainties are predicted and utilized to better compress the video in the entire framework.
>
> Both refinement nets are implemented with neural networks and jointly optimized with other modules in an end-to-end manner. We agree such technical details are important for reproducing our results and are an important part of our main contributions. We update our submission with more technical details, and we summarize them here to address the reviewer’s concern.
>
> Consider a model with 4-head decoders, the Prediction Refine Net takes four motion compensation (MC) predictions and the previous decoded reference frame as input and outputs four refined MC predictions. The Reconstruction Refine Net takes four reconstructions (sum of decoded residuals and refined MC predictions) and four refined MC predictions as input, and outputs one final refined reconstruction. It is important to retain the implicit uncertainty (as an ensemble of outputs) from the multi-head decoders to later stages, which is why the Prediction Refine Net must generate four refined MC predictions. Refining multiple inputs into one output resolves any potential uncertainty, and we perform such operation only once in the last module of whole framework.

---

> > ### Author Response · Authors · 2021-11-17
> > **Response to Reviewer BZ5c - Part 2**
> >
> > > Predictive uncertainties are analyzed by assuming the predictions from multi-head decoders follow the mixture of Gaussian model. I think the number of decoders should be sufficiently large to support this claim.
> >
> > We agree modeling the outputs of multi-head decoders with Gaussian mixture models (GMM) is a strong claim. As far as we know, only with neural tangent kernels in infinitely-wide neural networks can we assume the network output to follow certain distributions [8]. However, we argue that modeling the output of ensemble models with GMMs is a quite common practice and is widely adopted by previous works. We combine the model outputs as a uniformly-weighted mixture model given by $p(y\mid x) = 1/M \sum_{m=1}^M p_{\theta_m}(y \mid x, \theta_m)$. For classification problems, this corresponds to averaging the softmax probabilities as used in [1,2,3]. For regression problems, we take the average of the regressed outputs in each head [3].
> >
> > Further, we would like to note that modeling the outputs of multi-head decoders are only used to analyze the behavior of the deep ensembles, rather than reaching any theorems. The motivation of this work is that existing deep codecs suffer from inaccurate decoded MVs and residuals, which is due to the epistemic uncertainty and aleatoric uncertainty explained in Section 3.1. Showing that our proposed ensemble model works like a charm is desirable, but it is also important to prove that the multi-head decoders can capture the predictive uncertainty. This implies that our proposed method alleviates the problems in our observations and that our empirical results are consistent with our conjectures.
> >
> > > Check for typos, punctuations, and articles.
> >
> > We have fixed the typos and notations in the updated submission.
> >
> > > Dependency on the optical flow estimation method.
> >
> > We do not think multi-head decoders depend on specific optical flow estimation methods to be effective. In fact, we also experimented with different optical flow networks, such as PWCNet, and multi-head decoders work equally well. We were not able to produce reliable results using RAFT as a submodule of our framework due to the recurrent structure.
> >
> > All these networks are pre-trained on FlyingChairs [4] and FlyingThings [5] with groundtruth optical flows, and all three optical flow networks learn to produce particularly good results. However, in deep video compression, we must balance the rate-distortion trade-off, and current state-of-the-art optical flow estimation methods usually produce optical flows that are too accurate and take too many bits to encode. We believe most current SOTA optical flow networks should work well as the motion estimation network in various predictive coding-based deep video codecs, including DVC [6], scale-space flow [7], and ours.
> >
> > > Failure cases?
> >
> > Failure cases of decoded frames have been added in Appendix A.7.
> >
> > [1] Szegedy et al. “Going Deeper with Convolutions.” In CVPR, 2015.
> >
> > [2] Lee et al. “Why M Heads are Better than One: Training a Diverse Ensemble of Deep Networks.” In arXiv, 2015.
> >
> > [3] Lakshminarayanan et al. “Simple and Scalable Predictive Uncertainty Estimation using Deep Ensembles.” In NeurIPS, 2017.
> >
> > [4] Dosovitskiy et al. “FlowNet: Learning Optical Flow with Convolutional Networks.” In ICCV, 2015.
> >
> > [5] Mayer et al. “A large dataset to train convolutional networks for disparity, optical flow, and scene flow estimation.” In CVPR, 2016.
> >
> > [6] Lu et al. “An end-to-end learning framework for video compression.” In PAMI, 2020.
> >
> > [7] Agustsson et al. “Scale-space flow for end-to-end optimized video compression.” In CVPR, 2020.
> >
> > [8] He et al. "Bayesian Deep Ensembles via the Neural Tangent Kernel." In arXiv, 2020.

---

### Official Review · Reviewer_Fyqz · 2021-11-02

**Correctness:** 2
**Technical Novelty And Significance:** 2
**Empirical Novelty And Significance:** 2
**Recommendation:** 5
**Confidence:** 3

**Main Review:**

This paper is interesting and a good try as a deep learning method for video compression. The experimental results seem strong. The ablation study looks extensive. However, the reviewer has several concerns.

1, the motivation of multi-head, i.e., the ensemble, is not clear. Why ensemble? What is its benefit?
2, in the introduction, the authors claim that existing methods suffer from the inaccurate estimation of optical flow. However, the proposed method also employs motion estimation, how can the proposed method avoid this issue?
3, in the ablation study, multiple heads look useful. In this case, why not use more heads?
4, poor representation. For example, the figures are not self-contained. One needs to refer to different parts in the texts to understand the figure, such as MV, Res in Fig. 1. There are many wrong sentences with grammar issues. Some are given but they are definitely not all.
a, caption in figure 2, "are cannot be"
b, pp4, "it follows that the upper ... is given by"
c, pp5, MEMSE
d, pp4, Fig.9e should be Fig.2d?
f, pp5. Fig.9f should be Fig.2e?



**Summary Of The Paper:**

This paper aims to capture the predictive uncertainty and proposes an ensemble-based video compression approach. In the proposed approach, an ensemble-aware loss is also introduced to seek diversity among ensemble members. Experimental results show its performance.

**Summary Of The Review:**

In general, this paper solves a classic problem with deep learning, which is interesting. While there are several issues (see the main review section), which prevent the reviewer from scoring it high with the current status.

---

> ### Author Response · Authors · 2021-11-17
> **Response to Reviewer Fyqz**
>
> We thank the reviewer for the feedback, and we address the concerns below:
>
> > the motivation of multi-head, i.e., the ensemble, is not clear. Why ensemble? What is its benefit?
>
> **Our motivation is that existing predictive coding-based methods rely on inaccurate intermediate representations**, such as the decoded optical flows and the decoded residuals. Deterministic but erroneous representations are used for motion compensation and reconstruction, and inevitably propagate errors to later stages, leading to suboptimal performance.
>
> We argue that **the epistemic uncertainty in the motion estimation outputs and the aleatoric uncertainty introduced by the quantization operation are the source of such inaccurate representations** (see Section 3.1). Therefore, **we propose to keep an ensemble of such intermediate representations with multi-head decoders**. Rather than blindly relying on one deterministic decoded MV or residual, our model learns the implicit predictive uncertainty as the diversity between ensemble members (as depicted in Figure 2f), and learns to handle inaccurate MVs or residuals when outputs from different ensemble members disagree.
>
> The benefits of the multi-head decoders are supported by the following observations.
> 1. In Section 3.1, we investigate the source of the errors in the decoded MVs and the residuals and depict them in Figure 3d and 3e. After optimizing our model, the unsupervised predictive uncertainty depicted in Figure 3f agrees with our conjectures, which means our model can effectively learn the predictive uncertainty. This shows that multi-head decoders can alleviate the problems from our observations.
> 2. In Section 4.2 and 4.4, quantitative results show that multi-head decoders can bring significant bitrate savings, and our proposed model can significantly outperform previous state-of-the-art deep codecs (more than 20% bitrate saving over DVC Pro).
>
> > in the introduction, the authors claim that existing methods suffer from the inaccurate estimation of optical flow. However, the proposed method also employs motion estimation, how can the proposed method avoid this issue?
>
> Our motivation is that existing predictive coding-based methods use inaccurate intermediate representations, such as the decoded optical flows and the decoded residuals, for motion compensation and reconstruction. Deterministic but erroneous representations propagate errors to later stages and lead to suboptimal performance. In video compression, we must balance the trade-off between bitrate and distortion. **While highly accurate optical flows help to improve the quality of motion compensation predictions, they also take many more bits to encode.** **Instead of improving the motion estimation and predicting better MVs, we turn to the decoders and hope to decode more reliable MVs and residuals.**
>
> **How can multi-head decoders avoid this issue?** We argue that the epistemic uncertainty in the motion estimation outputs and the aleatoric uncertainty introduced by the quantization operation are the source of such inaccurate representations. Therefore, we propose to keep an ensemble of such intermediate representations, and help the network learn an implicit predictive uncertainty (as depicted in Figure 2f). By optimizing the framework in an end-to-end fashion, the network is aware of the uncertainty and inaccuracy in MVs and residuals of certain regions, and learns to handle such inconsistencies between different ensemble members.
>
> To address the reviewer’s concern, we would also like to clarify that **we are focusing on the errors in the decoded optical flows and the decoded residuals, rather than the errors in the estimated MVs from the motion estimation module or the residuals before coding**. Notice that the decoded ones are actually used for motion compensation and reconstruction while the outputs from the motion estimation module are not directly used for bilinear warping.
>
> > in the ablation study, multiple heads look useful. In this case, why not use more heads?
>
> When designing our model, we balance the trade-off between the model complexity and the final performance. As reported in the complexity analysis and the ablation study on the number of heads in Section 4.4, we investigated the complexity and the performance of models with different numbers of heads. Finally, we choose the model h=4 for our main experimental results with a significant performance improvement but a limited complexity increase.
>
> > poor representation. For example, the figures are not self-contained. One needs to refer to different parts in the texts to understand the figure, such as MV, Res in Fig. 1. There are many wrong sentences with grammar issues.
>
> We thank the reviewer for pointing out some grammar issues and errors in figure references. We apologize for these typos and have fixed these issues in the updated submission.

---

### Official Review · Reviewer_ZZDv · 2021-11-03

**Correctness:** 3
**Technical Novelty And Significance:** 2
**Empirical Novelty And Significance:** 3
**Recommendation:** 6
**Confidence:** 3

**Main Review:**

Strengths:
+ The paper is well-structured.
+ The proposed model outperforms previous state-of-the-art models of deep learning video codecs, and the ablation study proves the effectiveness of each module.

Weaknesses:
- There are quite some theoretical analysis in the paper to support the need of using ensemble/multi-head, which seems nice, but I am not sure if this is necessary. The idea of multi-head has been widely used in transformer models and it is well-known that this can significantly improve performance in practice. Can the authors better contrast their idea with the multi-head idea used in typical transformers?
- In Eq 10, what does i stand for? each pixel? Also, why take summation of j=1..h, isn't the min is over j which means at 1 single location, only 1 prediction is used for computing L2? Further it would be very helpful if the authors explain how this is designed beyond just citing Lee et al. Why this loss function can encourage diversity? Pls add this in the main text as this is one of the main contributions of this paper.
- For results in Fig 3, why the proposed method's curve stops earlier than other approaches? How would the method compare with other methods when bpp is high such as> 0.15 in (a), and can the proposed approach achieve the overall better PSNR when allowing larger bpp?
- Compared to traditional codecs, the deep learning compressors have large model size in general, so that hard to deploy on low-capability device. I assume this proposed approach would have similar issue?

**Summary Of The Paper:**

This paper works on end-to-end deep learning video compression. The authors study the inherent uncertainty and accordingly propose a so-called ensemble approach which is in effect multi-head decoder. A so-called ensemble-aware loss is proposed to encourage the diversity between ensemble members. Further, adversarial training is incorporated. Experiments show that the proposed model outperforms previous state-of-the-art models such as DVC Pro (Lu et al., 2020b) and Lu et al. (2020a), and the ablation study proves the effectiveness of each module.


**Summary Of The Review:**

I am on borderline as there are specific concerns outlined above regarding contribution of multi-head this simple idea, presentation of the ensemble-aware loss, results when bpp is high, etc. But I also believe deep codes is a promising direction and we shall encourage more work in this field and may go with a lower bar. If the authors can address my questions, I will lean to accept.

---

> ### Author Response · Authors · 2021-11-17
> **Response to Reviewer ZZDv - Part 1**
>
> We thank the reviewer for the feedback, and we address the concerns below:
>
> > There are quite some theoretical analysis in the paper to support the need of using ensemble/multi-head, which seems nice, but I am not sure if this is necessary. The idea of multi-head has been widely used in transformer models and it is well-known that this can significantly improve performance in practice. Can the authors better contrast their idea with the multi-head idea used in typical transformers?
>
> **As far as we know, the investigation in the multi-head attention in transformer models is limited.** We summarize some of them below:
> 1. Multi-head attention improves performance, as supported by most SOTA transformer models.
> 2. Heads in multi-head attention share common projections [5]. Some of them learn highly similar representations and can be pruned without impacting performance. [2,4].
> 3. Multi-head attention produces attention on different spaces, such as different linguistical roles in NLP [2]. This is quite different from multi-head decoders that are similar in function space but different in parameter space. They only perform differently and show large predictive uncertainty given out-of-distribution inputs.
> 4. The main advantage of multi-head attention may be training stability [3].
> 5. While there are connections between multi-head attention and deep ensembles, they are mostly conjectures since the similarity in formality rather than solid theoretical analysis.
>
> In fact, our theoretical analyses are built on the task of deep video compression and explain the motivation of introducing ensemble methods. **They prove the legitimacy of multi-head decoders, explain why multi-head decoders would work, and the analysis on predictive uncertainty shows that the behavior of multi-head decoders is consistent with our conjectures.** The motivation is that existing predictive coding-based methods decode inaccurate optical flows and residuals. We argue that the epistemic uncertainty from motion estimation and the aleatoric uncertainty from the quantization operation are the source of such errors. Thus, we propose to learn the predictive uncertainty by keeping an ensemble of intermediate representations with multi-head decoders. We found that the unsupervised predictive uncertainty learned by our model (in Fig 2(f)) is consistent with the visualization of the aleatoric and epistemic uncertainty in Fig 2(d) and 2(e). This indicates that our model truly learns predictive uncertainty and shows the efficacy of the multi-head decoders.
>
> > For results in Fig 3, why the proposed method's curve stops earlier than other approaches? How would the method compare with other methods when bpp is high such as> 0.15 in (a), and can the proposed approach achieve the overall better PSNR when allowing larger bpp?
>
> Our model is tested with the publicly available i-frame model, Cheng2020 [1]. Since the released models of Cheng2020 did not cover ranges with higher bitrates, we test our model with VTM [2] to address the reviewer’s concern and similar conclusion could be made at higher bitrate range. More details could be found in Appendix A.5 with RD curves of our proposed model and DVC Pro using Cheng2020 and VTM as I-frame models on HEVC Class B and UVG. Our model can outperform previous state-of-the-art models in higher bitrates and multi-head decoders can bring significant bitrate saving compared to DVC Pro using the same I-frame model.
>
> From the quantitative results in Section 4.2, we could see that the multi-head decoders bring significant bitrate savings mostly by reducing the bitrate while maintaining the same reconstruction quality. Besides training models with lambda=256, 512, 1024, 2048, we are currently training another model with lambda=4096 for higher bitrates. We believe this would help to show the advantage of our approach in higher bitrates.

---

> > ### Author Response · Authors · 2021-11-17
> > **Response to Reviewer ZZDv - Part 2**
> >
> > > In Eq 10, what does i stand for? each pixel? Also, why take summation of j=1..h, isn't the min is over j which means at 1 single location, only 1 prediction is used for computing L2? Further it would be very helpful if the authors explain how this is designed beyond just citing Lee et al. Why this loss function can encourage diversity? Pls add this in the main text as this is one of the main contributions of this paper.
> >
> > In Eq 10, i stands for each pixel location in the 2D lattice. At each pixel location, the loss is equal to the h times the minimum loss. However, **we format the loss in this way to show two important properties of our ensemble-aware loss: (i) the loss of each head is relaxed to the minimum loss to encourage disagreement (min over h), and (ii) at any iteration, every head is supervised to ensure the performance of individual head (sum over h).** Even when k=1, every prediction is used for computing L2 to properly update the weights in each head with back propagation. We revise the formality of our ensemble-aware loss in the updated submission. In the updated formula, t is the index of head from 1 to h, i and j are locations in the 2D lattice.
> >
> > We thank the reviewer for pointing out that Section 3.3 focused on the motivation and benefits of our proposed loss and might not have explained how this loss is designed clearly enough. We update Section 3.3 with more details. We summarize the idea here to address the reviewer’s concern.
> >
> > * **Why we need a new ensemble-aware loss?** Simply training with the rate-distortion loss leads to the degeneration of the ensemble model where every head learns similar representations and yield comparable performance with the single-model counterpart. If we adopt the oracle set-loss in [6], each head only sees a portion of all data and both the performance of individual heads and the whole model is suboptimal.
> > * **How does our proposed ensemble-aware loss work?** We propose an ensemble-aware loss that (i) allow diversity between heads by relaxing the magnitude of the loss, and (ii) ensure performance of each head by updating every head at every iteration. Since the gradients propagated to each head depend on the head with the minimal MSE, rather than the MSE of the head itself, we can effectively induce randomness and therefore encourage diversity. This loss can be easily implemented in PyTorch by forwarding the gradients before and after the clamp operation.
> > * **Why is our ensemble-aware loss better?** Our proposed ensemble-aware loss is a simple yet effective method to induce diversity between ensemble members. **The hyperparameter k allows us to balance the trade-off between diversity and individual performance, and can gracefully adapt to different number of ensemble members or different tasks.** With the rise of multi-head models and deep ensembles, we believe this is an important contribution to the community.
> >
> > > Compared to traditional codecs, the deep learning compressors have large model size in general, so that hard to deploy on low-capability device. I assume this proposed approach would have similar issue?
> >
> > Yes, we agree in general the current deep video codecs have large model size and computational complexity, which prevents us from deploying deep learning-based decoders to mobile devices. However, we would like to emphasize that deep learning-based video compression is still at its early age and the compression community are focusing on improving the performance of such methods rather than resolving the practical concerns of their applications. Recently, we have seen that deep learning-based video codecs start to outperform competitive traditional codes on various benchmark datasets, and we expect to see more theoretical and empirical research outcomes soon.
> >
> > [1] Cheng et al. “Learned Image Compression with Discretized Gaussian Mixture Likelihoods and Attention Modules.” In CVPR, 2020.
> >
> > [2] VTM reference software for VVC. https://vcgit.hhi.fraunhofer.de/jvet/VVCSoftware_VTM.
> >
> > [3] Voita et al. " Analyzing Multi-Head Self-Attention: Specialized Heads Do the Heavy Lifting, the Rest Can Be Pruned.” In ACL, 2019.
> >
> > [3] Liu et al. " Multi-head or Single-head? An Empirical Comparison for Transformer Training.” In arXiv, 2021.
> >
> > [4] Michel et al. " Are Sixteen Heads Really Better than One?” In NeurIPS, 2019.
> >
> > [5] Cordonnier et al. "Multi-Head Attention: Collaborate Instead of Concatenate." In arXiv, 2020.
> >
> > [6] Lee et al. "Why M Heads are Better than One: Training a Diverse Ensemble of Deep Networks". In arXiv, 2015.

---

> > > ### Author Response · Authors · 2021-11-22
> > > **Response to Reviewer ZZDv - Updated Results**
> > >
> > > We thank the reviewer for the feedback. We would like to update our response with a new result.
> > >
> > > **Regarding comment 3: "why the proposed method's curve stops earlier than other approaches?":** We've finished training our model with lambda=4096 and have updated the results in Appendix A.5. As we could see, our model significantly outperforms previous state-of-the-art methods over all bitrates. Further, multi-head decoders can bring significant bitrate savings compared to DVC Pro regardless of the I-frame model used.

---

### Official Review · Reviewer_3SEn · 2021-11-08

**Correctness:** 3
**Technical Novelty And Significance:** 2
**Empirical Novelty And Significance:** 2
**Recommendation:** 5
**Confidence:** 4

**Main Review:**

(+) Experimental results suggest that the ensemble and multihead based proposed method outperforms prior works.
(-) Novelty of the paper is not clear. It seems that the general framework of the proposed compression scheme follows DVC but with multihead mv decoder based on equally-weighted Gaussian mixture.
(-) The ensemble-based idea does not seem original as there have been several prior networks (Lakshmnarayanan et al, Agustsson et al. etc) as alluded by the authors. The difference between the proposed and prior work should be concisely and explicitly described.
(-) Readability should improve with the notation used in the text (section 3.1) correlates with the figure 1.
(-) Should make a performance comparison with the recent paper which is readily available on archive
Missing reference
Hao Chen, Bo He, Hanyu Wang, Yixuan Ren, Ser-Nam Lim, Abhinav Shrivastava, NeRV: Neural Representations for Videos NeurIPS 2021


**Summary Of The Paper:**

This paper proposes an ensemble-based video compression model to capture the predictive uncertainty of intermediate predictions. A loss is constructed to encourage diversity between ensemble members, and the paper investigates the benefit of incorporating adversarial training in video compression. The experimental result shows that the proposed model can save more than 20% compared to DVC Pro.

**Summary Of The Review:**

It is difficult to pinpoint the novelty of the proposed compression algorithm as it stitches various ideas from various areas (ensemble, multi-head, FGSM) that have already been proposed.

---

> ### Author Response · Authors · 2021-11-17
> **Response to Reviewer 3SEn - Part 1**
>
> We thank the reviewer for the feedback, and we address the concerns below:
>
> > Novelty of the paper is not clear. It seems that the general framework of the proposed compression scheme follows DVC but with multihead mv decoder based on equally-weighted Gaussian mixture.
>
> **Predictive coding-based framework consists of motion compensated prediction and residual coding. It is widely used by traditional codecs and many SOTA learning-based video codecs.** While there are many other approaches based on 3D auto-encoders [1,2] and neural representations [3], motion compensation is still one of the simplest yet effective methods to exploit temporal correlation. DVC is the typical predictive coding-based deep video codec that implements each module with a neural network. It is simple and extendable, and is widely used as a baseline by [4,5,6]. It is therefore a natural idea to follow the predictive coding-based framework and use DVC as the baseline framework.
>
> Based on DVC, **we propose multi-head decoders to learn an ensemble of intermediate representations** observing that existing methods suffer from inaccurate but deterministic optical flows and residuals. **The legitimacy of this method is supported by our theoretical results in the aleatoric and epistemic uncertainty in deep video compression and the visualization of the unsupervised predictive uncertainty (Section 3.2)**. We also propose **a novel ensemble-aware loss (Section 3.3)** that can significantly improve the performance of deep ensembles with various numbers of ensemble members. Finally, we prove the efficacy of our method by showing **significant performance improvement (more than 20% bitrate saving) on benchmark datasets (Section 4.2)**.
>
> > The ensemble-based idea does not seem original as there have been several prior networks (Lakshmnarayanan et al, Agustsson et al. etc) as alluded by the authors. The difference between the proposed and prior work should be concisely and explicitly described.
>
> **How is our model different from previous deep ensembles?** Almost all, if not all, previous deep ensembles [7,8,9,10] implement deep ensembles as stand-alone neural networks and the benefits of deep ensembles are trivial when the model size and computational complexity easily increases by several hundred percent. In addition, all models except [7] showed their results in controlled experiments without proving the efficacy on benchmark datasets. Instead, our multi-head decoders are lightweight (+4% in model size) and outperform previous SOTA methods on benchmark datasets.
>
> Why can multi-head decoders work so well? Firstly, **we ensemble intermediate layers of the model rather than the whole network itself.** Secondly, **the idea of deep ensembles in deep video compression is related to the uncertainty introduced in the quantization operation in lossy entropy coding.** Quantization operations in compression introduce unwanted uncertainty when decoding the frames. We explain this novel observation in Section 3.1 and visualization of the unsupervised predictive uncertainty supports our theoretical insights and shows the efficacy of our model.
>
> **How is our model different from Agustsson et al. [4]?** [4] proposed to regress a scale field as the Gaussian varaince and use Gaussian blurring when MVs are not estimated well by motion estimation. We summarize the differences between our work and [4] as follows:
> 1. **Motivation is different.** [4] focused on MVs that are not estimated well by the motion estimation. We focus on the aleatoric and epistemic uncertainty in the decoded optical flows and residuals (see Section 3.1).
> 2. **Approach is different.** [4] use Gaussian blurring to reduce the negative effects of inaccurate optical flows. Our multi-head decoders maintain an ensemble of outputs and process them in parallel.
> 3. **Regressing variance (sigma in Gaussian blur) is not reliable.** Regressing classification confidence or variance as the scale field in [4] is an ill-posed question [11]. For out-of-distribution data, the model cannot estimate MVs well, and at the same time, the scale field will not reflect the quality of the MV. We cannot rely on unstable scale fields to properly blur the frames.
> 4. **Our work is built on solid theoretical insights.** [4] adopted a quite engineering approach, Gaussian blurring, to deal with inaccurate MVs. Multi-head decoders are built on theoretical insights (see Section 3.1) and learn to process an ensemble of outputs. Instead, deep ensemble is a perfect choice to capture such uncertainty.
> 5. **Our method is theoretically more general, extendable, and achieves much higher performance.** We show in Section 3.2 that our method learns a more adaptive and more general representation of MVs, and the quantitative results of our method are also superior.

---

> > ### Author Response · Authors · 2021-11-17
> > **Response to Reviewer 3SEn - Part 2**
> >
> > > Readability should improve with the notation used in the text (section 3.1) correlates with the figure 1.
> >
> > We thank the reviewer for pointing out the inconsistencies in the notations. The notations are fixed and can be found in the updated submission.
> >
> > > Should make a performance comparison with the recent paper which is readily available on archive Missing reference Hao Chen, Bo He, Hanyu Wang, Yixuan Ren, Ser-Nam Lim, Abhinav Shrivastava, NeRV: Neural Representations for Videos NeurIPS 2021.
> >
> > We thank the reviewer for pointing us towards this related work that follows a very different approach. We notice that the arXiv version of this paper was not published until Oct 26, 2021, which is why we did not compare with NeRV before the submission deadline in early October. We have updated our submission by including this work in the literature review and adding NeRV as a comparison in the quantitative results section.
> >
> > [1] Pessoa et al. “End-to-End Learning of Video Compression using Spatio-Temporal Autoencoders.” In SiPS, 2020.
> >
> > [2] Habibian et al. “Video Compression With Rate-Distortion Autoencoders.” In ICCV, 2019.
> >
> > [3] Chen et al. “NeRV: Neural Representations for Videos.” In NeurIPS, 2021.
> >
> > [4] Agustsson et al. “Scale-space flow for end-to-end optimized video compression.” In CVPR, 2020.
> >
> > [5] Hu et al. “Improving Deep Video Compression by Resolution-adaptive Flow Coding.” In ECCV, 2020.
> >
> > [6] Lu et al. “Content Adaptive and Error Propagation Aware Deep Video Compression.” In ECCV, 2020.
> >
> > [7] Szegedy et al. “Going Deeper with Convolutions.” In CVPR, 2015.
> >
> > [8] Lakshminarayanan et al. “Simple and Scalable Predictive Uncertainty Estimation using Deep Ensembles.” In NeurIPS, 2017.
> >
> > [9] Lee et al. “Why M Heads are Better than One: Training a Diverse Ensemble of Deep Networks.” In arXiv, 2015.
> >
> > [10] Wang et al. “Ensemble learning-based rate-distortion optimization for end-to-end image compression.” In TCSVT, 2020.
> >
> > [11] Yarin Gal. "Uncertainty in Deep Learning." PhD Thesis.

---

### Author Response · Authors · 2021-11-17
**Response to All Reviewers**

First, we would like to thank all reviewers for their efforts in reading and reviewing our submission. We genuinely appreciate the reviewers’ detailed comments, questions, and constructive suggestions. We update our submission with the following modifications:
* We add more technical details on the implementation of multi-head decoders in Appendix A.1.
* We update Section 3.3 with more details on ensemble-aware losses, including how general ensemble-aware losses induce diversity, design details of our novel ensemble-aware loss, and why is it better.
* We add quantitative results of our model using VTM as the i-frame model to show the performance of our model under higher bitrates in Appendix A.5.
* We compare our model with more recently published works (after the ICLR submission deadline), such as NeRV from NeurIPS 2021, in terms of the method we use (in Section 2) and the quantitative performance (in Section 4.2).
* We add both good and bad qualitative results of decoded frames in Appendix A.7.
* We fix typos, notations, and figure references.

We would also like to address some of the common concerns here:

> What’s the main theoretical and empirical contribution of this work?

Intermediate representations in predictive coding-based codecs, such as MVs and residuals, are usually inaccurate and errors are propagated to later stages, leading to sub-optimal performance. We identify the sources of such errors: (i) epistemic uncertainty in motion estimation and (ii) aleatoric uncertainty from the quantization operation in lossy entropy coding. Based on the observations, we propose multi-head decoders and learn an ensemble of the intermediate representations. We further propose a novel ensemble-aware loss that can effectively improve the performance of deep ensembles. Visualization of unsupervised predictive uncertainty shows the legitimacy of multi-head encoders, and experimental results show that our proposed model achieves more than 20% bitrate saving than previous SOTA method DVC Pro.

> How are multi-head decoders different from previous deep ensembles and scale-space flow by Agustssons et al.?

**How are multi-head decoders different from previous ensembles?**
1. Previous deep ensembles train multiple stand-alone networks while our model learns an ensemble of intermediate layers. As shown in our quantitative results and ablation study, our multi-head decoders are light-weight yet quite effective.
2. Our multi-head decoders are designed for deep video codecs and can effectively capture the uncertainty introduced from the lossy entropy coding.
3. Most previous results on deep ensembles are shown in controlled experiments while our method achieves promising results on benchmark datasets.

**How are multi-head decoders different from scale-space flow proposed in Agustsson et al.?**
1. **Motivation is different.** Agustsson et al. noticed that some optical flows cannot be estimated well by the motion estimation module. We focus on uncertainty in the decoded optical flows and residuals, including the aleatoric uncertainty introduced by the quantization operation and the epistemic uncertainty in motion estimation.
2. **Approach is different.** Agustsson et al. use Gaussian blurring to reduce the negative effects of inaccurate optical flows. Our multi-head decoders maintain an ensemble of outputs and process them in parallel.
3. **Regressing variance (sigma in Gaussian blur) is not reliable.** Regressing classification confidence or variance as the scale field in Agustsson et al. is an ill-posed question [1]. For out-of-distribution data, the model cannot estimate MVs well, and at the same time, the scale field will not accurately reflect the quality of the MV. We cannot rely on unstable scale fields to properly blur the frames.
4. **Our work is built on solid theoretical insights.** Rather than adopting a quite engineering approach, Gaussian blurring, our multi-head decoders are built on the theoretical insights in Section 3.1 and supported well by the unsupervised predictive uncertainty.
5. **Our method is theoretically more general, extendable, and achieves much higher performance.** We show in Section 3.2 that our method learns a more adaptive and more general representation of MVs and the quantitative results of our method is also superior.

[1] G. Yarin. "Uncertainty in Deep Learning."

---

### Decision · Program_Chairs · 2022-01-20

**Decision:**

Reject

**Comment:**

This paper exposes a method for video compression based on multi-head models.
The reviewers seem to agree that the results are interesting, and worth publishing.
On the other hand, there are many concerns raised on the quality of the writing, with grammatical mistakes and confusing parts. The motivation for the multi-head models, as well as its novelty, has been questioned in all reviews. Although the authors rebuttal has lead some reviewers to increase their score, it's still very concerning that authors needed to explain the main point of the paper to each reviewers. I think that the authors should polish this paper, taking into account the reviewers feedback, which would make a stronger paper, and submit it again in a future venue. I therefore recommend reject for this paper.